# Challenging Safety and Efficacy of Retinal Gene Therapies by Retinogenesis

**DOI:** 10.3390/ijms22115767

**Published:** 2021-05-28

**Authors:** Elena Marrocco, Rosa Maritato, Salvatore Botta, Marianna Esposito, Enrico Maria Surace

**Affiliations:** 1Telethon Institute of Genetics and Medicine, 80078 Napoli, Italy; marrocco@tigem.it (E.M.); m.esposito@tigem.it (M.E.); 2Department of Translational Medicine, University of Naples Federico II, 80131 Naples, Italy; rosa.maritato@unina.it; 3Department of Genetics and Development, Columbia University Irving Medical Center, New York, NY 10032, USA; sb4158@cumc.columbia.edu

**Keywords:** gene therapy, adeno-associated virus (AAV), retinal degeneration, transcription, zinc finger, retinitis pigmentosa, rhodopsin, autosomal dominant

## Abstract

Gene-expression programs modulated by transcription factors (TFs) mediate key developmental events. Here, we show that the synthetic transcriptional repressor (TR; ZF6-DB), designed to treat Rhodopsin-mediated autosomal dominant retinitis pigmentosa (RHO-adRP), does not perturb murine retinal development, while maintaining its ability to block Rho expression transcriptionally. To express ZF6-DB into the developing retina, we pursued two approaches, (i) the retinal delivery (somatic expression) of ZF6-DB by Adeno-associated virus (AAV) vector (AAV-ZF6-DB) gene transfer during retinogenesis and (ii) the generation of a transgenic mouse (germ-line transmission, TR-ZF6-DB). Somatic and transgenic expression of ZF6-DB during retinogenesis does not affect retinal function of wild-type mice. The P347S mouse model of RHO-adRP, subretinally injected with AAV-ZF6-DB, or crossed with TR-ZF6-DB or shows retinal morphological and functional recovery. We propose the use of developmental transitions as an effective mode to challenge the safety of retinal gene therapies operating at genome, transcriptional, and transcript levels.

## 1. Introduction 

Broadly categorized, gene therapies (GTs) for inherited monogenic disorders, are based on the use of nucleic acid-based targeted to the causal mutated genes or to their downstream effectors (mutation independent approaches) [1]. Gene-targeted approaches for loss of function mutations of the gene product (typically recessive disorders) aim to correct the underlying cause of the genetic defects providing the functional copy of the defective gene by gene augmentation (also termed gene replacement, supplementation, substitution, or addition). Instead, gene targeted inactivation methods are used to disable the toxic effects of monoallelic gain-of-function mutations (dominant disorders). Alternatively, gene-mutation independent approaches lay the ground for targeting relevant pathways secondarily affected by the mutated causal gene.

The variety of currently known inherited retinal diseases, the knowledge of their distinct underlying mutated gene involved, and the advantageous experimental characteristics, made the eye attractive to test a multiplicity of groundbreaking therapeutic strategies [2,3]. The milestone achievement of Adeno-associated virus (AAV) vector RPE65 gene augmentation therapy for Laber Congenital Amaurosis type 2 (LCA2) [4], promoted a number of clinical tests based on gene supplementation therapies for retinal recessive disorders [5,6]. Until recently, treatments of retinal gain-of-function mutations lagged behind. Thus, retinal dominant disorders remain an urgent unmet medical need. Yet, recently a series of innovative approaches have been conceived and they are becoming increasingly more effective experimentally [7]. 

Accounting for 20–30% of all RP cases and with more than 150 dominant mutations described, autosomal dominant retinitis pigmentosa (adRP), due to *RHODOPSIN* gene (RHO-AdRP, RP4 (OMIM: 613731)) [8,9], has become a prominent disease model to assess the effects of Rho targeted and Rho mutation independent approaches [10]. The latter include the use of pharmacological small molecules, chaperones, and histone deacetylase inhibitors, nutritional supplements, cell death inhibitors, and neurotrophic factors [11]. Recently, we also introduced the use of a Rho mutation independent approach based on the use of a microRNA [12]. Instead, RHO targeted interventions aim to disable the Rho gene at genome, transcription, and transcript level. Clustered Regularly Interspaced Short Palindromic Repeats (CRISPR)-based therapies acts at genomic level [13]. Cas9 endonucleases activity induces Rho silencing by targeted mutagenesis (by non-homologous-end-joining (NHEJ) mechanism) at the Rho locus [14,15,16,17,18,19]. Ribozymes [20], RNA interference [21,22], and antisense oligonucleotides degrade Rho mRNA transcript upon pairing [23]. Finally, the transcriptional repressor (TR)-based paradigm we introduced, blocks transcription by binding the proximal promoter regulatory region of Rho. We showed that natural transcription factor (TF) KLF15 ectopically expressed in rod photoreceptors [24], the synthetic TF ZF-KRAB [25], or the DNA-binding protein ZF6-DB [26], allow efficient on-target blockade of RHO transcription. 

Therefore, differentially from gene augmentation approaches in which a gene is administered to add the function of the one inactivated by the mutation, RHO targeted therapeutic approaches actively operate endogenously at genome, transcriptional, and transcript levels. Thus, if dysregulated they may alter the cellular homeostasis and the natural course of the disease. In particular, the mechanisms of action of these systems are inherently prone to off targeting, thus potentially turning on or off unintended and harmful pathways [27,28]. Therefore, ways to assess functionally off targeting effects are hugely needed. 

Along development dynamic changes of the transcriptome and the cellular epigenome landscape ultimately driven by TFs activity, enable the transitions of multipotent progenitor cells from proliferation to cell-specific differentiation. The mouse retinal progenitors proliferate, exit the cell cycle, and differentiate in a well-established overlapped patterned fashion, in prenatal and early postnatal life [29,30,31,32,33]. Here, we investigate whether the TR DNA-binding protein ZF6-DB, which operates similarly to TFs, interferes with retinal development changes from a safety standpoint and whether an early therapeutic intervention before the onset of the Rho-adRP disease presentation result in a better clinical benefit than a treatment administered at later time points. 

## 2. Results and Discussion

### 2.1. Early Postnatal AAV-Mediated Somatic Delivery of ZF6-DB Did Not Interfere with Retinal Development in WT Animal and Resulted in Morphological and Functional Benefit in P347S adRP Mice

The six main retinal cell types (rod, cone, horizontal, bipolar, amacrine, ganglion cells [34] arise from multipotent neural progenitors. In the mouse retina, waves of linage-specific proliferations and differentiation start from embryonic day 10 (E10) and end 10 days after birth (Postnatal day 10, P10) [30,31]. Therefore, at any given stage between E10 and P10, the developing retina is a mixture of proliferating retinal progenitor cells, newly postmitotic committed cells, and differentiating cells. Rods, which represent 80% of the total number of murine retinal cells (6.4 million per retina) [35], reach the plateau of cells born (exit from cell cycle) around P1, whereas Rhodopsin onset of expression occurs about P5 [31]. To determine whether ZF6-DB impact these retina developmental transitions, we subretinally administered an AAV8 [36] carrying ZF6-DB under the CMV promoter elements (AAV8-CMV-ZF6-DB, Figure 1A) at P4 in WT animals. At P30 we measured the light responses of ZF6-DB injected mice by Electroretinogram (ERG) analysis, which showed no differences compared to sham injected controls (a- and b-waves in scotopic condition; Figure 1B,C; photopic responses, Appendix A). We next evaluated the impact of early delivery (P4) of ZF6-DB in the P347S adRP mouse model and compared the effect to a later stage of AAV-ZF6-DB delivery (P14). P347S transgenic mice, carrying the RHO dominant mutation P347S, develop a severe retinal degeneration, modelling RHO-AdRP [37]. These transgenic mice harbor the entire transcription unit of the RHO allele including both the proximal promoter and the 3′ downstream flanking regions [37]. Therefore, as previously shown, they enabled the assessment of RHO transgene repression by three distinct TRs after AAV vectors subretinal delivery [24,25,26]. Those studies demonstrated that the TRs can discriminate between the proximal promoter of human transgenic RHO carrying the P347S mutation and the endogenous murine Rho copies, by selectively binding to a specific regulatory region of the human RHO promoter, which is not present in the murine Rho promoter regulatory elements. ERG analysis showed that the group of P4 ZF6-DB injected animals resulted in highly significant preservation of retinal responses compared to sham injected controls (a- and b-waves in scotopic condition; Figure 1D,E; photopic responses, Appendix A). In addition, P4 delivery in P347S mice resulted in higher responses than P14 treated retina at any light intensities in scotopic condition (Figure 1D,E). This result was also qualitatively confirmed by the analysis of the shapes of the ERG wave light evoked responses (Figure 1F). Furthermore, co-immunofluorescence analysis at P30 of animals treated with AAV-ZF6-DB, revealed that ZF6-DB (anti-HA tag) localized in the photoreceptor nuclei (ZF6-DB expression appropriately localized toward the periphery of rod photoreceptor nuclei) [38] of the outer nuclear layer (ONL), and Rhodopsin in the outer segment (OS) of rods. Contralateral control retina showed profound thinning of ONL nuclei and retention of Rhodopsin in the remaining cell soma rod photoreceptor (Figure 1G). 

### 2.2. Generation of a Transgenic Mouse Expressing ZF6-DB in the Retina (TR-ZF6-DB)

To express the RHO ZF6-DB in the mouse retina by germ-line transmission, we generated a construct containing the proximal promoter region of the human Guanine Nucleotide Binding Protein1 (GNAT1) gene upstream of the transcriptional repressor ZF6-DB (Figure 2A). This 564-long promoter GNAT1 fragment containing the 5′ UTR is able upon Adeno-associated virus (AAV) vector subretinal delivery to express the EGFP selectively in murine rod photoreceptors [26], and data are not shown. In genotype positive animals for the TR-ZF6-DB transgenes (Methods), ZF6-DB transgene expression was assessed in the retina at postnatal day 10 (P10). As shown in Figure 2B, we found positive expression of the transgene (TR-ZF6-DB) by RT-PCR in several pups, although weak, confirming germ-line transmission and retinal-specific expression. To determine whether the expression of the TR-ZF6-DB resulted in modification of retina function, we performed ERGs. The ERGs analysis showed no differences in light-evoked responses between TR-ZF6-DB and wild-type (WT) one month old animals (a- and b-waves in scotopic condition; Figure 2C,D; photopic responses, Appendix A). To further confirm the absence of apparent changes in the TR-ZF6-DB retina we performed immunofluorescence analysis with Rhodopsin (Figure 2E) and glia-derived neurotrophic factor (GFAP; Figure 2F), a marker of both Müller cells and astrocytes activation upon inflammation and retinal cellular stress. Rho expression pattern was found similar between TR-ZF6-DB and WT animals and GFAP reactivity, virtually absent with only rare and scattered positive cells in the ganglion cell layer [39] (Appendix A, reactive GFAP positive control). Immunostaining with anti-HA tag did not show a detectable expression of the ZF6-DB (Appendix A), suggesting that the ZF6-DB transgene expression is below the threshold of sensitivity. These results suggest that the ZF6-DB expression in the retina did not affect retinal development and retinal function and morphology short-term in adult animals.

### 2.3. Double Transgenic Mice Expressing Both the ZF6-DB and the Human Mutant RHO P347S Allele

To assess whether TR-ZF6-DB represses human RHO-specific P347S mutant allele, we crossed the TR-ZF6-DB transgenic lines with P347S mice to obtained TR-ZF6-DBxP347S double transgenic animals. RT-PCR studies showed that in the double transgenic animals the expression levels of the ZF6-DB transgene were weak, however significant (Figure 3A). In addition, we measured the levels of expression of the human P347S RHO allele compared to the Rho murine alleles, as well as mouse cone Arrestin 1 (mArr1) as control, in both double transgenic (TR-ZF6-DBxP347S) and in the P347S transgenic RHO-AdRP model. As shown in Figure 3B, the activity of ZF6-DB significantly reduced the human RHO allele. Furthermore, ERG analysis showed that in double transgenic (TR-ZF6-DBxP347S) mice retinal activity was preserved compared to P347S controls, both in scotopic and photopic conditions (Figure 3C,D and Appendix A). This functional protection was mirrored at morphological level. Immunofluorescence analysis with anti-RHO antibody demonstrate a thicker outer nuclear layer (ONL) of double transgenic (TR-ZF6-DBxP347S) mice compared to P347S controls (Figure 3F), which in addition retained Rhodopsin in the ONL (Figure 3E). Consistently with quantitative expression data observed in pigs [26], and by others [40], these results support that low levels of ZF6-DB expression enable retinal protection from RHO-adRP disease progression.

## 3. Summary and Conclusions

Developmental transitions are controlled by transcription factors (TFs), which control transcriptional and epigenetic dynamics underpinning progenitor cells proliferation and cell-specific identity [33,41]. Therefore, it dawned on us that retinal gene therapies based on transcriptional modulators, which mimic TF mode of action, may be appropriately challenged to assess their safety and specificity by developmental retinal transitions. We found that the expression of ZF6-DB along retinogenesis by early postnatal retinal somatic delivery or by germline transmission did not affect short-term retinal function in wild-type animals. In addition, in a severe disease model of *RHO-ADRP*, both somatic and germline early ZF6-DB expression resulted in highly significant preservation of retinal structure, improvement of trafficking and localization of Rhodopsin, and ultimately in protection of retinal function. 

We have previously described AAV vector delivery properties in the murine fetal retina as early as embryonic day 13 [42]. Furthermore, we also demonstrated that in utero delivery of AAV-RPE65, mediated the correction of the RPE65 biochemical and functional defects, establishing the safety and efficacy of gene transfer in the developing retina [43]. These studies underscored the beneficial effect of gene therapy in preventing the onset and progression of retinal degenerative disorders, showing that timely early intervention leads to the best clinical results [44]. Yet, the mode of action of gene augmentation operates by adding a functional gene into a cellular system in which that gene is not functional, instead, gene therapy targeted approaches designed for the treatment of the gain of function mutations, acting at genomic, transcript, and transcriptional levels, may perturb the cellular homeostasis due to an unbalanced-on target and/or off target effects. Here we introduced the use of murine retinogenesis as an indirect mode to evaluate safety of a therapeutic acting at transcriptional level.

## 4. Materials and Methods

### 4.1. Plasmid Construction and Generation of Transgenic Mice TR-ZF6-DB

The hGNAT1-ZF6-DB was generated by cloning the ZF6-DB transgene in the pAAV2.1 hGNAT1-hRHO plasmid [26] by removing the hRHO transgene and replace it with ZF6-DB, using NotI and HindIII restriction enzymes. The resulting construct was linearized with NheI and XhoI restriction enzymes. The linearized DNA was purified with Kit Midi Prep (Qiagen, Hilden, Germany) and used for microinjection into fertilized CB6 (female BALB/c, male C57Bl/6) eggs (Plaisant SRL, Roma, Italy). Mouse-tail biopsy were used to extract genomic DNA for genotyping with 5′_CTCCCAATTATGCCCTCTCA_3′ (hGNAT1Fw) and 5′_ACCGCCCTTCTTATTCTGGT_3′ (ZF6-DB-Rev) primers. Six male and 8 female positive founders (F0) have been crossed one another, 2 F0 male and female couples provided litters with positive retinal expression of the ZF6-DB transgene (Section 4.4). 

### 4.2. Animal Studies

All procedures were performed in accordance with institutional guidelines for animal research and all the animal studies were approved. Mice were housed and bred at the animal facility of Biotechnology Centre of the Cardarelli Hospital (Naples, Italy) and maintained under a 12 h light/dark cycle. P347S transgenic mice were maintained as F0 by crossing them with themselves and were crossed with C57BL/6J mice (Charles Rivers Laboratories, Calco, Co, Italy) to generate experimental F1 mice, which received AAV subretinal injection (as described [19,21]). TR-ZF6-DB transgenic mice were maintained as F0 by crossing them with themselves. To generate double transgenic mice (TR-ZF6-DB-P347S), TR-ZF6-DB F0 transgenic mice were bred with P347S F0 mice.

### 4.3. AAV Vector Preparation and Subretinal Administration

AAV vectors were produced by the Telethon Institute of Genetics and Medicine (TIGEM) AAV Vector Core, by triple transfection of HEK293 cells, followed by 2 rounds of CsCl2 purification [45]. For each viral preparation, physical titers (genome copies, gc/mL) were determined by averaging the titer achieved by dot-blot analysis and by PCR quantification using TaqMan (Applied Biosystems, Waltham, MA, USA). Intraperitoneal injection of ketamine and medetomidine (100 mg/kg and 0.25 mg/kg, respectively), then AAV vectors were delivered subretinally via a trans-scleral transchoroidal approach as described by [46]. Once anesthetized, the pupils are dilated with 0.5% tropicamide. A conjunctival incision is made and used to grasp and rotate the ocular globe. Subsequently, a sclerotomy is made by penetrating the sclera with a 27-gauge needle. A 33-gauge needle connected to a Hamilton syringe is inserted through the sclerotomy 2 mm posterior to the temporal limbus. The cannula is then advanced tangential to the curvature of the globe to the subretinal space in the posterior pole. Half microliter in post-natal day 4 mice (corresponding to a dose of 0.5 × 10^9^ AAV vector genomes) or one microliter in P14 and adult mice of solution (corresponding to a dose of 1 × 10^9^ AAV vector genomes) containing purified recombinant adeno-associated virus is injected into the subretinal space with the assistant pushing the plunger. The detachment of the retina typically covers ~1/4 fraction of the retina.

### 4.4. qReal Time PCR

RNAs from retina were isolated using RNAeasy Mini Kit (Qiagen, Hilden, Germany), according to the manufacturer’s protocol. cDNA was amplified from 1 µg isolated RNA using QuantiTect Reverse Transcription Kit (Qiagen), as indicated in the manufacturer’s instructions. Transcript levels of mice retina were measured by quantitative Real-Time PCR using the LightCycler 480 (Roche) and the following primers: ZF6_forward (GGCAAGAGCTTTAGCCAGAA) and ZF6_reverse (ACAGGCGTGCTGTTTCTTTT), mRho_Forward (GACTCTGCCAGCTTTCTTTGCT) and mRho_ Reverse (GCGTCGTCATCTCCCAGTGGA), hRho_Forward (CCATCCCAGCGTTCTTTGCC) and hRho_Reverse (CCTCATCGTCACCCAGTGGG). All the reactions were standardized against murine Actβ and Gapdh using the following primers: mAct_Forward (CAAGAT- CATTGCTCCTCCTGA) and mAct_reverse (CATGCTACTCCTGCTTGCTGA), mGapdh_forward (GTCGGTGTGAACGGATTTG) and mGapdh_reverse (CAATGAAGGGGTCGTTGATG).

### 4.5. Immunostaining

Eyes were harvested after euthanasia and fixed in 4% paraformaldehyde (PFA) in Phosphate-buffered saline (PBS) O.N. The following day the eyes were dehydrated in 30% sucrose in PBS until they sunk and then embedded and frozen in optimal cutting temperature compound (OCT). The cry-sections were obtained by cryostat sectioning, 12 µm thick.

Rhodopsin straining: Frozen retinal sections were washed once with PBS and then fixed for 10 min in 4% PFA. Frozen retinal sections were permeabilized with 0.1% Triton X-100, rinsed in PBS, blocked in 20% normal goat serum (NGS), and then incubated overnight at 4 °C in a mouse anti-1D4 rhodopsin antibody (Abcam, Cambridge, MA, USA) diluted 1:500 in 10% NGS. After three rinses with 0.1 M PBS, sections were incubated in goat anti-mouse IgG conjugated with Alexa Fluor 488 (anti-mouse 1:1000, Molecular Probes, Invitrogen, Carlsbad, CA, USA) for 1 h followed by another three rinses with PBS. Vectashield (Vector Lab Inc., Peterborough, UK) was used to visualize nuclei. 

HA staining: Sections were immersed in a retrieval solution (0.01 M citrate buffer, pH 6.0) and boiled three times in a microwave. After the blocking solution (10% FBS, 10% NGS, 1% BSA) was added for 1 h. The primary antibody mouse anti-HA (1:300, Covance) was diluted in a blocking solution and incubated overnight at 4 °C. The secondary antibody (Alexa Fluor 594, anti-mouse 1:1000, Molecular Probes, Invitrogen, Carlsbad, CA, USA) has been incubated for 1 h. 

GFAP staining: Anti-glial fibrillary acidic protein (GFAP; 1:400; Z0334; Dako Agilent, Santa Clara, CA, USA) staining were performed for 1 h in 0.3% Triton/4% normal goat serum then incubated overnight at 4C in 0.1% Triton/2% normal goat serum. The secondary antibody (Alexa Fluor 488, anti-rabbit 1:1000, Molecular Probes, Invitrogen, Carlsbad, CA, USA) has been incubated for 1 h. 

To quantify the thickness of the retina in TR-ZF6-DBxP347S double transgenic mice and P347S mice, histological reconstruction of the superior and inferior hemiretina was performed with IMAGEJ using 3 animals/8 sections per animal, 100 µm apart for each point starting from the optic nerve head (ONH), “spidergraph form” as reported [47].

### 4.6. Electrophysiological Testing

The method used was described previously [12,13]. Briefly, mice were dark reared for 3 h and anesthetized by intraperitoneal injection of ketamine and medetomidine (100 mg/kg and 0.25 mg/kg, respectively). Flash ERGs were evoked by 10 ms light flashes generated through a Ganzfeld stimulator (Costruzione Strumenti Oftalmici, Florence, Italy) and registered as previously described [26]. 

ERG analysis in scotopic conditions the responses evoked by 5 stimuli (from −4 to +1.3 log cd⋅s/m^2^) with an interval of 0.6 log unit were delivered. To minimize the noise, 3 ERG responses were averaged at each 0.6 log unit stimulus from −4 to 0.0 log cd⋅s/m^2^, while one ERG response was considered for higher (0.0 to +1.3 log cd⋅s/m^2^) stimuli. The time interval between stimuli was 10 s from −5.4 to 0.7 log cd⋅s/m^2^, 30 s from 0.7 to +1 log cd⋅s/m^2^, or 120 s from +1 to +1.3 log cd⋅s/m^2^. Wave amplitudes recorded in scotopic conditions were plotted as a function of increasing light intensity (from −4 to +1.3 log cd⋅s/m^2^). The photopic ERG was recorded after the scotopic session by stimulating the eye with ten 10 ms flashes of 20.0 cd⋅s/m^2^ over a constant background illumination of 50 cd⋅s/m^2^. 

### 4.7. Statistical Analysis

ERG data were analyzed with GraphPad Prism 9.1 software (Northside Dr.Suite 560, San Diego, CA, USA). Statistical significances were determined using t-tests corrected for multiple comparisons with the Holm–Sidak method (α = 0.05). Each light intensity was analyzed individually; individual variance for each group were assumed. Significances are indicated as follows: * *p* < 0.05, ** *p* < 0.01, *** *p* < 0.001. ERG results are displayed as box-and-whisker plots (boxes show the 25% and 75% quantile range, whiskers indicate the 5% and 95% quantiles spanning the interquartile range, line within box represents the median). 

Spidergram quantification was analyzed with GraphPad Prism 9.1 software (Northside Dr.Suite 560, San Diego, CA, USA). Statistical significances were determined using *t*-tests corrected for multiple comparisons with the Holm–Sidak method (α = 0.05). Individual variance for each row were assumed. Significances are indicated as follows: * *p* < 0.05, ** *p* < 0.01, *** *p* < 0.001.

The Mann–Whitney test with median and interquartile range was used to compare groups (qReal time PCR). All tests are 2-tailed. A *p*-value of <0.05 was considered significant.

## Figures and Tables

**Figure 1 ijms-22-05767-f001:**
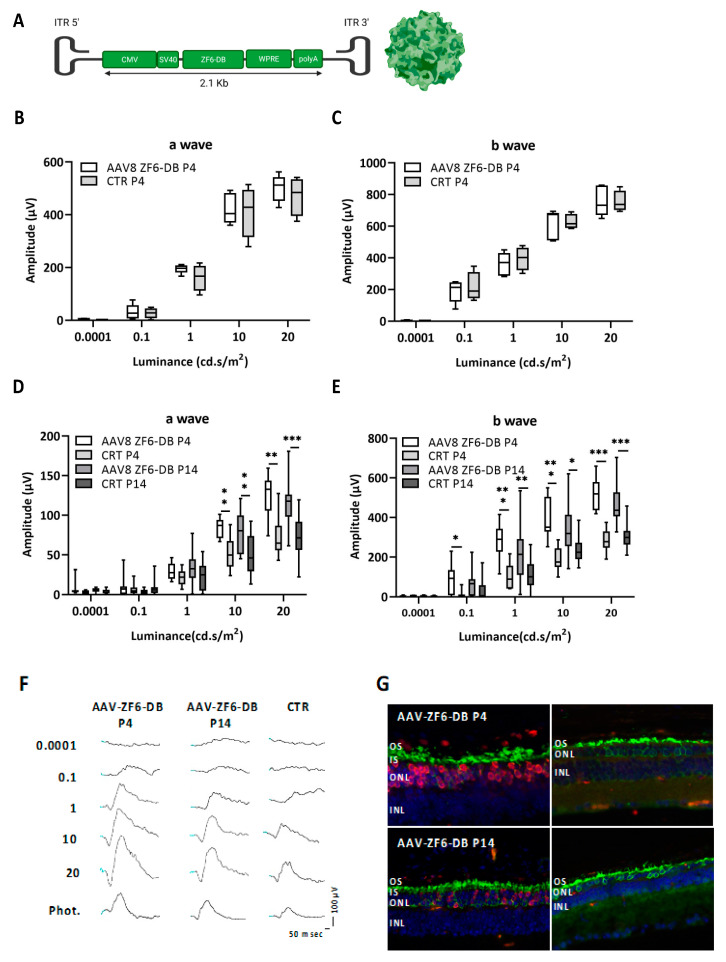
Early somatic delivery of ZF6-DB transcriptional repressor in the WT and P347S adRP murine model by adeno-associated virus (AAV) vector. (**A**) AAV8-CMV-ZF6-DB construct representation showing the cytomegaloviruses (CMV) promoter, including the SV40intron, the woodchuck hepatitis posttranscriptional regulatory element (WPRE), and the bovine growth hormone polyA (bGH). (**B**,**C**) a- and b-waves amplitudes recorded in scotopic conditions plotted as a function of light intensity (log cd * s/m^2^) in one-month old WT animals treated at postnatal day 4 (P4, white box; *n* = 5; controls CTR grey box; *n* = 4). (**D**,**E**) a- and b-waves in one-month old P347S adRP mice treated with AAV8-CMV-ZF6-DB at postnatal day 4 (P4, white box; *n* = 9) and P14 (grey box; *n* = 18), compared with sham injected contralateral controls eyes (grey box; *n* = 9, P4 and *n* = 18, P14); Statistics: *t*-test corrected with Holm–Sidak method for multiple comparisons. * *p* ≤ 0.05, ** *p* ≤ 0.01, *** *p* ≤ 0.001. (**F**) Representative ERG track responses in scotopic and photopic conditions in one-month old P347S adRP mice treated with AAV8-CMV-ZF6-DB at postnatal day 4 and P14 compared with sham injected contralateral controls eyes. (**G**) Representative cryo-retinal sections co-immunostained with Rhodopsin (green) and ZF6-DB HA-tag (red) antibodies in one month old P347S mice treated at P4 and P14 and contralateral controls, sections were counterstained with Vectashield to visualize nuclei. OS, Outer Segment. IS, Inner Segment. ONL, Outer nuclear Layer. INL, Inner Nuclear Layer.

**Figure 2 ijms-22-05767-f002:**
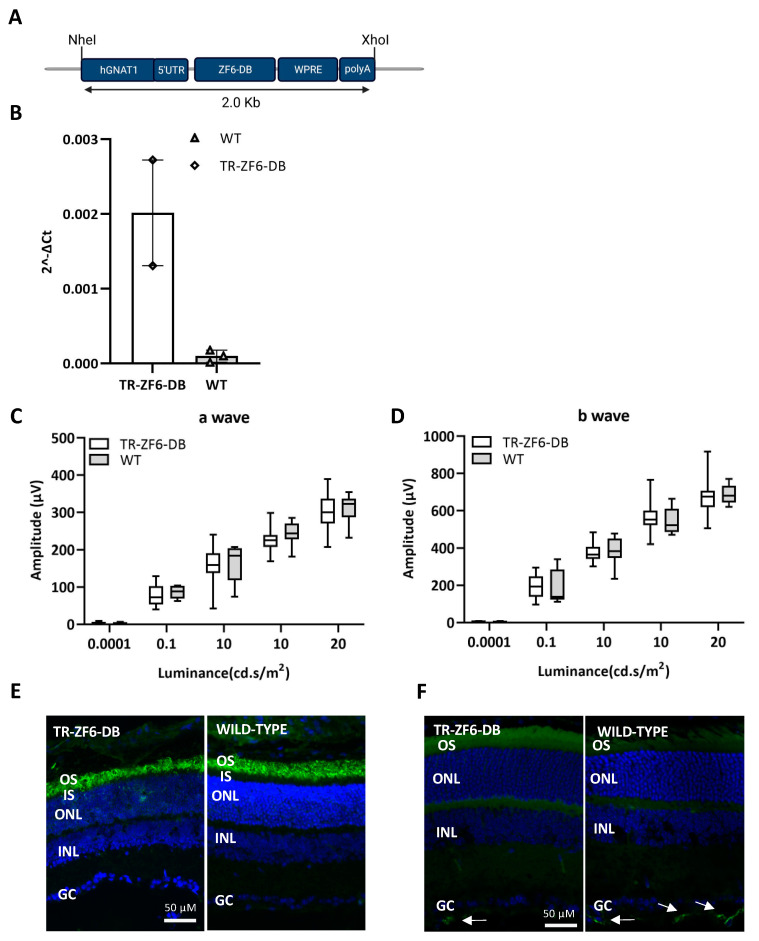
Transgenic mice expressing the ZF6-DB (TR-ZF6-DB) transcriptional repressor in the murine retina. (**A**) representation of TR-ZF6-DB transgenic construct showing the human Guanine Nucleotide Binding Protein1 (hGNAT1) promoter fragment including the 5′ UTR cloned upstream the ZF6-DB coding sequence followed by the WPRE and the bovine growth hormone polyA (bGH). (**B**) qReal Time PCR of mRNA expression levels (2^−∆CT^) of the ZF6-DB transgene on postnatal day 10 (P10) retina compared to wild-type match controls. TR-ZF6-DB, *n* = 4 (2 littermate pups belonging to two independent TR-ZF6-DB F0 litter (4 pulled retina); WT *n* = 6 (2 littermate pups (4 pulled retina) from the same litter). (**C**,**D**) a- and b-waves amplitudes recorded in scotopic conditions plotted as a function of light intensity (log cd * s/m^2^) in one-month old TR-ZF6-DB (*n* = 4, white box) and WT controls (*n* = 4, controls CTR grey box). Statistics: *t*-test corrected with Holm–Sidak method for multiple comparisons. (**E**) Rho immunofluorescence (green) histological analysis of one-month old TR-ZF6-DB and WT controls. (**F**) Glia-derived neurotrophic factor (GFAP) immunofluorescence (green) histological analysis of one-month old TR-ZF6-DB and WT controls. Arrows indicate positive in the ganglion cell layer, GC. OS, Outer Segment. IS, Inner Segment. ONL, Outer nuclear Layer. INL, Inner Nuclear Layer.

**Figure 3 ijms-22-05767-f003:**
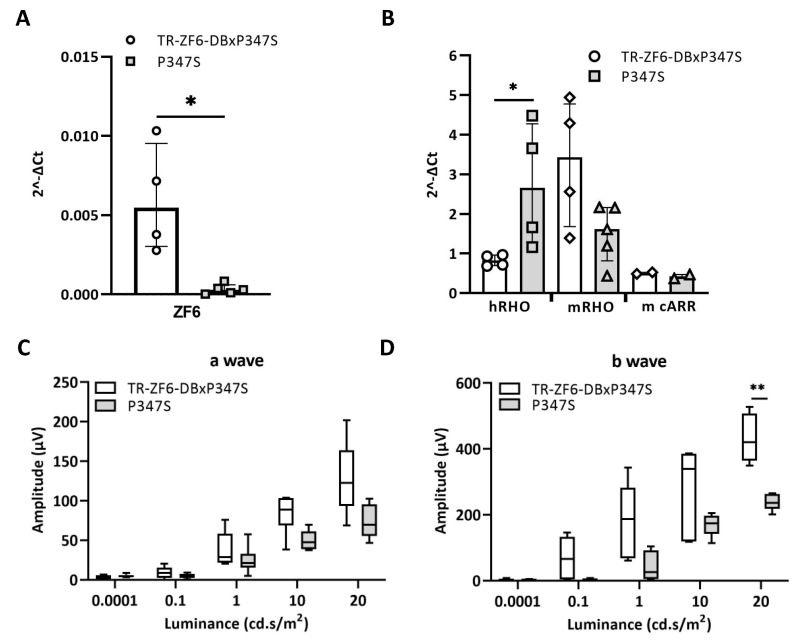
Double transgenic animals TR-ZF6-DBxP347S, derived by the cross of TR-ZF6-DB with P347S RHO-AdRP mice. (**A**) qReal Time PCR of mRNA expression levels (2^−∆CT^) of the ZF6-DB in double TR-ZF6-DBxP347S (*n* = 4 eyes) compared to P347S RHO-AdRP (*n* = 5 eyes) on P30 retina. (**B**) qReal Time PCR of mRNA expression levels (2^−∆CT^) of murine endogenous alleles (mRho), human P347S RHODOPSIN mutant allele, and mouse cone Arrestin 1 (mArr1) in P30 TR-ZF6-DB-P347S and P347S RHO-AdRP mice (on the same animals of B). (**C**,**D**) a- and b-waves amplitudes recorded in scotopic conditions plotted as a function of light intensity (log cd * s/m^2^) in one-month old TR-ZF6/DB-P347S (*n* = 5, white box) and P347S RHO-AdRP mice (*n* = 5, grey box). Statistics: *t*-test corrected with Holm–Sidak method for multiple comparisons. * *p* ≤ 0.05, ** *p* ≤ 0.01, *** *p* ≤ 0.001. (**E**) Rho Immunofluorescence (green) histological analysis of one-month old TR-ZF6/DB-P347S and P347S RHO-AdRP mice. OS, Outer Segment. IS, Inner Segment. ONL, Outer nuclear Layer. INL, Inner Nuclear Layer. GC, ganglion cell layer. (**F**) Retinal “spidergram” showing ONL thickness from the optic nerve head (ONH) to the inferior and superior hemispheres of double TR-ZF6-DBxP347S (*n* = 3 eyes) and P347S (*n* = 3 eyes). Statistics: *t*-test corrected with Holm–Sidak method for multiple comparisons. * *p* ≤ 0.05, ** *p* ≤ 0.01, *** *p* ≤ 0.001.

## Data Availability

Not applicable.

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
