# Peer review of "Challenging Safety and Efficacy of Retinal Gene Therapies by Retinogenesis"

_ijms, 2021, doi:10.3390/ijms22115767_

Round 1

Reviewer 1 Report

The authors presented a potential AAV-gene therapy for RHO-adRP and evaluated the potential retinal toxicity of the transgene. This is a very interesting approach. However, the manuscript shows only a limited amount of data.  It is also crucial to substantiate the data and clarify how some experiments were performed, and the number of animals used.

Major:

- Figures are poorly labelled.

  • ERG data are not properly presented. The data should be presented using a box-and-whisker plot, and show the 25 and 75% quantile range, whiskers indicate the 5 and 95% quantiles and the asterisks indicate the median of the data.
  • Fig1: Please add the data related to the scotopic a-wave; present the photopic data in another graphic. On D, replace "," by ".". On E, please label the retinal layers and add the scale bar.
  • In section 2.2, one-month-old animals do not show any obvious "deleterious" effect from the transgene expression by ERG or by analysing RHO expression. Two points can be raised, the authors should look to later time points (e.g. 3M and 5M), and detailed morphological analysis might be done (e.g. GFAP, number of Photoreceptors, microglia activation, etc). With the data presented the authors cannot exclude a long-term "toxicity".
  • Figure 2: It is not clear how many samples/animals were used. If for B the n=2, then statistics cannot be performed. There is also no mention of how many animals were used on C. Please add the scale bar on D. It is also important to clarify in which cells are the transcript expressed. The author could perform an in situ to demonstrate the presence of it.
  • Line 184: the authors mentioned a "thicker ONL" in the double mutant animals, however, no quantification is provided. Please support the statement by measuring the thickness of the retina at different distances of the ONH and presented using a spidergram. 
  • Figure 2: A, is the "n" the number of different animals? Please clarify. How many animals were recorded in C. Please add a scale bar on D.
  • Material and methods do refer to too many other studies. Once there is no space limitation please provide the reader with the complete information.

Minor:

 - Line 120: Please add "cell soma" before rod photoreceptor.

  • Section 2.2 - The authors are referring to the wrong figure, it should be Fig 2 and not Fig 1.
  • Line 155: Please add the "data not shown" to the manuscript. 
  • Line 220-221: It is unclear what the authors want to say in the last sentence.
  • Line 243: how subretinal injections were performed? How the confirmation of it was done? And how much of the retina was transduced?
  • Line 252: Please mention the dose and volume of AAV used.
  • Line 256: Much likely 1ug and not 1mg was used.
  • Line 268: How the eyes were prepared? Please add the reference of the 1D4 antibody.
  • Line 284: How the animals were anaesthetized.
  • Please add a statistical section to the material and methods.

Author Response

We thank the reviewer for the thoughtful comments and suggestions to the manuscript.

Please find below the point-by-point response. The revised manuscript has changed accordingly.

The authors presented a potential AAV-gene therapy for RHO-adRP and evaluated the potential retinal toxicity of the transgene. This is a very interesting approach. However, the manuscript shows only a limited amount of data.  It is also crucial to substantiate the data and clarify how some experiments were performed, and the number of animals used.

Major:

- Figures are poorly labelled.

We revised the figures throughout.

  • ERG data are not properly presented. The data should be presented using a box-and-whisker plot, and show the 25 and 75% quantile range, whiskers indicate the 5 and 95% quantiles and the asterisks indicate the median of the data.

In the revised manuscript we plotted the ERG data using a box-and-whisker plot as suggested.

  • Fig1: Please add the data related to the scotopic a-wave; present the photopic data in another graphic. On D, replace "," by ".". On E, please label the retinal layers and add the scale bar.

The a-wave and the photopic data are shown as suggested. The retinal layers are labelled, and the scale bar added.

  • In section 2.2, one-month-old animals do not show any obvious "deleterious" effect from the transgene expression by ERG or by analysing RHO expression. Two points can be raised, the authors should look to later time points (e.g. 3M and 5M), and detailed morphological analysis might be done (e.g. GFAP, number of Photoreceptors, microglia activation, etc). With the data presented the authors cannot exclude a long-term "toxicity".

We agree with the reviewer that the long-term claim cannot be sustained and accordingly we stated that the results are “short-term”. In the revised version of the manuscript, we added the GFAP immunofluorescence analysis, which showed no reactivity in Transgenic retina (Figure 2F).

  • Figure 2: It is not clear how many samples/animals were used. If for B the n=2, then statistics cannot be performed. There is also no mention of how many animals were used on C. Please add the scale bar on D. It is also important to clarify in which cells are the transcript expressed. The author could perform an in situ to demonstrate the presence of it.

We clarified in the Figure 2 legend the number of animals used. We agree that the identification of the transgene would have been important. However, we did not have the chance to perform in situ hybridization.

  • Line 184: the authors mentioned a "thicker ONL" in the double mutant animals, however, no quantification is provided. Please support the statement by measuring the thickness of the retina at different distances of the ONH and presented using a spidergram. 

Accordingly, we quantified the thickness of the retina relative to the distance from the ONH and presented the data using the “spidergram”.

  • Figure 2: A, is the "n" the number of different animals? Please clarify. How many animals were recorded in C. Please add a scale bar on D.

The number of animals is added to the revised manuscript, as well as the scale bar.

  • Material and methods do refer to too many other studies. Once there is no space limitation please provide the reader with the complete information.

As requested, we completed the Material and methods to provide more completed information

Minor:

 - Line 120: Please add "cell soma" before rod photoreceptor. Added

  • Section 2.2 - The authors are referring to the wrong figure, it should be Fig 2 and not Fig 1. Corrected
  • Line 155: Please add the "data not shown" to the manuscript. Added
  • Line 220-221: It is unclear what the authors want to say in the last sentence. Sentence changed.
  • Line 243: how subretinal injections were performed? How the confirmation of it was done? And how much of the retina was transduced? Answered
  • Line 252: Please mention the dose and volume of AAV used. Added
  • Line 256: Much likely 1ug and not 1mg was used. Corrected
  • Line 268: How the eyes were prepared? Please add the reference of the 1D4 antibody. Added
  • Line 284: How the animals were anaesthetized. Added
  • Please add a statistical section to the material and methods. Added

Reviewer 2 Report

The manuscript is interesting, but there are some significant carelessness errors and shortcomings that the authors definitely need to correct before publishing.

1. The Materials and Methods section does not include the relevant section of Statistical Analysis, where should be reflected: 
  a) the normality check of the data to be analyzed, 
  b) relevant and correct between-group comparison tests, 
  c) applied central tendency and dispersion criteria (either arithmetic mean with SD or median with IQR).

2. There are two "Figure 2" (Figure 3 is missing) and there is no reference to Figure 2 in the text of the manuscript.

3. In the text should refer to all Figure parts (A, B, C, D).

4. Although the captions below the figures mention that  the two-tailed Student's t test was applied, it has been not correctly applied in all cases, as Figure 2B and Figure 3A, B show significant SD differences between the compared groups and thus, the applied Student's t test (as the parametric method) for data analysis was not the correct methodological approach. Instead the non-parametric Mann-Whitney test with median and interquartile range should be used.

5. Definitely in all figures (Fig. 1 - A, B, C, D, E; Fig. 2, 3 - A, B, C, D) it is necessary to increase the font size of axes and their designations, including all textual notations and explanations, because in this manuscript version it is difficult to see and read what's on graphs (images).

6. Review text and correct formatting errors

Author Response

We thank the reviewer for the thoughtful comments and suggestions to the manuscript.

Please find below the point-by-point response. The revised manuscript has changed accordingly.

The manuscript is interesting, but there are some significant carelessness errors and shortcomings that the authors definitely need to correct before publishing.

  1. The Materials and Methods section does not include the relevant section of Statistical Analysis, where should be reflected: 
    a) the normality check of the data to be analyzed, 
    b) relevant and correct between-group comparison tests, 
      c) applied central tendency and dispersion criteria (either arithmetic mean with SD or median with IQR).

The section of Statistical Analysis has been added to the revised manuscript

  1. There are two "Figure 2" (Figure 3 is missing) and there is no reference to Figure 2 in the text of the manuscript.

Corrected accordingly

  1. In the text should refer to all Figure parts (A, B, C, D).

Corrected accordingly

  1. Although the captions below the figures mention that the two-tailed Student's t test was applied, it has been not correctly applied in all cases, as Figure 2B and Figure 3A, B show significant SD differences between the compared groups and thus, the applied Student's t test (as the parametric method) for data analysis was not the correct methodological approach. Instead the non-parametric Mann-Whitney test with median and interquartile range should be used.

As pointed out by the reviewer we corrected the statistical method with the Mann-Whitney test.

  1. Definitely in all figures (Fig. 1 - A, B, C, D, E; Fig. 2, 3 - A, B, C, D) it is necessary to increase the font size of axes and their designations, including all textual notations and explanations, because in this manuscript version it is difficult to see and read what's on graphs (images).

Size increased

  1. Review text and correct formatting errors

Corrected

Round 2

Reviewer 1 Report

The manuscript is now clearly improved. However, a small issue still remains, the GFAP staining on FIG2 F seems to not work properly, even when fibres crossing the retina are not observed the Muller glial cell endfeet should be clearly visible. On the figure only a high level of background staining is present. 

Author Response

We thank the reviewer for the thoughtful comment and suggestion to the manuscript.

The manuscript is now clearly improved. However, a small issue still remains, the GFAP staining on FIG2 F seems to not work properly, even when fibres crossing the retina are not observed the Muller glial cell endfeet should be clearly visible. On the figure only a high level of background staining is present. 

We changed Figure 2F. Currently, the Figure shows lower levels of background and scattered GFAP positivity in ganglion cell layer.

Reviewer 2 Report

The corrections were made in accordance with the recommendations and are satisfactory.

Author Response

Thank you